# Copper-Based Volumetric Filler Dedicated for Ag Paste for Depositing the Front Electrodes by Printing on Solar Si Cells

**DOI:** 10.3390/ma11122493

**Published:** 2018-12-07

**Authors:** Małgorzata Musztyfaga-Staszuk, Grzegorz Putynkowski, Robert Socha, Maciej Stodolny, Piotr Panek

**Affiliations:** 1Welding Department, Silesian University of Technology, Konarskiego 18A, 44-100 Gliwice, Poland; 2Centrum Badań i Rozwoju Technologii dla Przemysłu S.A. (Research and Development Center of Technology for Industry) Złota 59, 00-120 Warsaw, Poland; office@cbrtp.pl; 3Institute of Catalysis and Surface Chemistry PAS, Niezapominajek 8, 30-239 Krakow, Poland; ncsocha@cyf-kr.edu.pl; 4ECN part of TNO, Solar Energy Westerduinweg 3, 1755 LE Petten, The Netherlands; maciej.stodolny@tno.nl; 5Institute of Metallurgy and Materials Science PAS. Reymonta 25, 30-059 Krakow, Poland; p.panek@imim.pl

**Keywords:** copper-based volumetric filler, front electrode, crystalline silicon solar cells

## Abstract

In this work we present research results on a new paste NPCuXX (where: NP—new paste, CuXX—component, XX—a modifier consisting of Ni and other important elements) based on a copper composite (CuXX) for fabrication of front electrodes in silicon solar cells. The CuXX composite is obtained by chemical processing of copper powder particles and can be used in two ways: as an additive to commercially available paste or as a base material for a new paste, NPCuXX. The CuXX offers the possibility to exchange up to 30 and 50 wt.% Ag into Cu, which significantly decreases the solar cells material costs, and therefore, the overall solar cell price. Emphasis was placed on a proper mass suitable fabrication process of the CuXX component. The NPCuXX paste has been applied both to conventional cell structures such as aluminum-back surface field (Al-BSF) and passivated emitter and rear contact (PERC), and finally solar cells with front electrodes deposited by screen-printing method were fabricated and characterized by current-voltage techniques. This paper reports the first implementation of the copper volumetric material into a screen print paste used in a high-temperature metallization process to fabricate the front contacts of Si solar cells with a highest fill factor of 77.92 and 77.69% for the abovementioned structures, respectively.

## 1. Introduction

The global market requirement for crystalline silicon (c-Si) solar cell production forces a cost reduction per unit power which increases the effectiveness of solar systems [1]. The effectiveness can be achieved either by further enhancing the energy conversion efficiency or by lowering the fabrication costs that requires the development of new production methods and materials [2]. The important and cost-intensive step in the manufacturing of the crystalline silicon solar cells is the contact formation [3]. The replacement of silver by one hundred times cheaper copper is presently one of the challenges for research in the photovoltaics (PV) sector [4]. 

According to the report ITRPV (International Technology Roadmap for Photovoltaics) 2017 [5], silver will continue to play an important role in the metallization production of cells based on crystalline silicon. In 2009, 300 mg of silver paste per cell was used, in 2016 it was already 100 mg per cell, what represented ~8% of the cell fabrication costs. As shown in Figure 1, it is anticipated that the reduction of Ag will continue through the replacement of silver with copper or other material. It is predicted that in 2027 the amount of silver used per cell will be 40 mg.

In 2016, a finger width of less than 50 μm was achieved. Precision of accurate printing in the line has improved to <10 μm (Figure 2a). The possibility of Ag reduction necessitated an increase in the number of bus bars up to four, while the use of a front metallization consisting of three bus bars was abandoned (Figure 2b). In the classification of photovoltaic metallization production methods, it is possible to distinguish mainly those used only for the front metallization, back, and bifacial. As shown in the report ITRPV 2017, PREC technology is gaining about 20% of market share, where on the other hand the Al-BSF share is shrinking. Moreover, Si-tandem technology is under development and silicon heterojunction (SHJ) technology shows a slowly increasing share [5]. The price of silver ranged from 5.58 to 14.22 USD/oz for the period 1999–2018 [6]. The biggest recorded increase in the price of the rebound was the value 46.47 USD/oz on 27 March 2011. The above data shows a slight price drop for silver in the long run. As of 17 August 2018, the price of silver is 14.06 USD/oz, which means that in the future it should also continue to drop [6]. Fluctuations in silver prices in recent years mean that the cost of cell metallization can be 4 Euro cents for 1 W_p_ or even 8 Euro cents for 1 W_p_. The high sensitivity of metallizing paste cost to silver price changes is a serious problem for cell producers and PV modules [7]. Crystalline Si (c-Si) module manufacturing costs concerning materials are the following [8]: crucibles—3%, graphite—2%, argon—1%, slurry—10%, wires—14%, wet chemical—3%, Ag paste—14%, Al/Ag paste—3%, Al paste—7%, others—3%, glass—7%, EVA (Ethylene Vinyl Acetale)—11%, backsheet—5%, frames—11%, and box—6%. The inability to estimate production costs can effectively inhibit investment in increasing production potential [9]. As shown in Figure 2, it is predicted that further efforts are being made to reduce the coverage of the front side of the solar cell with metallization. This enables Ag reduction by means of implementing more bus bars, thus reducing resistive losses in the fingers.

In summary, it can be concluded that we are still at the beginning of PV market development, and according to the market data presented [10], as of 28 October 2015, the average price for multi-crystalline cells is 1.403 USD/oz and monocrystalline 1.583 USD/oz. 

As a result of the work presented in this paper, it is planned to introduce a copper-based volumetric filler paste on the market, which will be about 55% cheaper than its counterparts. Importantly, a lower cost of the paste will contribute to a reduction in the price of a monocrystalline cells by 4.5%, and multicrystalline cells by 5% while maintaining their quality parameters. All technologies that will accelerate this process will be supported with interest from cell manufacturers [11].

The proposed method of limiting Ag share in the contact paste by the authors of the work is innovative on a global scale. The work being done is a research work, and the results presented in it have not been achieved by any other scientific and research unit in the world.

## 2. Fabrication Process of the CuXX Component and Paste

The CuXX component was prepared by chemical method using as starting material—a commercially available copper powder (CNPC-FB00, CNPC Powder Group Co., Shanghai, China) of average grain diameter of 1 µm. The powder was selected on the basis of industrial availability. The applied powder showed uniform size distribution that can be observed in Figure 3. The powder was covered by an XX barrier layer with the use of an electroless deposition method from aqueous plating bath. The deposition method utilized reduction of nickel (II) salt by phosphorus species in the presence of surfactants. The application of surfactants in the deposition was necessary because the powder showed hydrophobic properties. In this process, the copper grains were suspended in the bath, the suspension was intensively stirred then the reducing agent was added to suspension. The resulting copper grains covered with the XX layer were rinsed thoroughly with deionized water and finally dried in air. The details of the deposition method are described elsewhere [12]. The dried CuXX powder was sifted through a 625 mesh sieve. The resultant CuXX component grains were covered with a compact layer containing nickel as a main barrier layer component. The deposition was elaborated in the semi-industrial dimension and finally fabricated on a batch of grains weighing 1000 g, which confirms the industrial applicability of the method and can be easily resized to a volume of 100 kg/h. The grade of coverage of the copper nickel surface is illustrated by the elemental distribution on the intersection line of the selected small CuXX grain (Figure 4).

The observed nickel coating is complete even on the surface of small grains (about 150 nm in the SEM picture), which confirms that the deposition method can be used for powders with a wide distribution of grain diameters. Moreover, the analysis of the crystallographic structure of the CuXX grain (Figure 5) showed that the barrier layer XX forms a suitable interface with the surface of the copper grain. The resulting CuXX component was used to make the NPCuXX paste.

The analysis showed a homogeneous distribution of the XX modifier on the Cu surface. The CuXX grain separated from the obtained CuXX powder has a homogeneous crystal structure, which is confirmed by the high-resolution analysis of the grain cross-section made by the TEM method (Figure 5). The homogeneity of the crystalline structure of the barrier layer XX is important from the point of view of material stability and behavior during the high-temperature metallization process of cell contacts. The deposition method is economically cost-effective, which in connection with the properties of the product allows it to be used for mass industrial production.

The choice of the chemical composition of the paste was investigated experimentally. A new type of paste was made with the participation of selected granules of the component using a carbonyl acetate (OKB) as a carrier and a thinner recommended by the manufacturer of the Du Pont pastes called Texanol. The mass and quantity of selected components were precisely determined and weighed using a laboratory scale, and then the mixture was prepared using a mechanical mixer. The ingredients were mixed for 5 h at a rotation of 60 rpm in room temperature in order to obtain a homogeneous paste. Then, paste rolling tests were carried out at various gaps between rolls of 60, 40, 20, and 5 μm, respectively. Each time after rolling, the size of the agglomerates in the paste was tested on the gradiometer. Carbonyl acetate was also used as a thinner. 

Although the exact compositions of the metallization pastes are kept as an industrial secret, it is possible to present the overall composition of a universal front metallization paste as it is presented in Table 1. The characteristics and parameters of the new paste NPCuXX with a new CuXX component are presented in Table 2.

The comparison between these two tables shows that the new NPCuXX paste is not inferior in quality to commercial pastes commonly used on the market, actually both pastes have very similar parameters.

In the first works on the development of the paste [20], results were presented in which the cells had an fill factor (FF) of 73%. Subsequent work concerned the study of the parameters of CuXX component molecules and the effect of NPCuXX paste on cell parameters [25,26]. The method of refining the paste and ingredients as well as the creation of links on the ECN ’line made it possible to achieve such spectacular results.

Figure 6 presents the pastes’ microstructure prior to screen printing, drying, and high temperature metallization process. Maps of superficial distribution of elements from the area are presented in Figure 6a,d,g. It was also found that morphologies of experimental paste A show a huge uniformity with a commercial Ag paste, which is connected with occurrences of numerous agglomerates in globule shape with diversified sizes from a few to tens of micrometers (Figure 6a,g). A bigger difference in the appearance of the paste was observed between experimental B and commercial Ag paste. This difference shows some heterogeneity (Figure 6d), which results from powder grains melting.

## 3. Material and Experiment 

The research work related to the implementation of the NPCuXX component and paste was carried out in three research centers in Poland, while the production of solar cells and the measurement of their parameters were carried out at the ECN part of TNO institute in the Netherlands. The main steps of the experiment are presented in the following paragraphs:

**Cells:** The ECN part of TNO provided PERC and Al-BSF cells with only the front side metallization.
p-type mono Cz-Si material,front side featured a POCl_3_ based ^n+^ emitter and an 80 nm SiN_x_ thick coating,rear side featured laser patterned AlO_x_/SiN_x_ dielectric and a local area Al-BSF including Ag soldering pads (only PERC concept),quantity of 110 cells was prepared: 10 reference cells (standard front Ag PV20A paste), 50 cells for metallization with NPCuXX paste A (including 25 cells for firing optimization), 50 cells for metallization with NPCuXX paste B (including 25 cells for firing optimization).

Copper-based volumetric filler paste: CBRTP prepared and delivered three types of paste: both A and B based on the CuXX component and commercial Ag PV20A Du Pont paste. The latter was marked as reference (REF). The A and B pastes differed from each other in the preparation, composition, and manufacturing procedure of the CuXX component. The pastes A and B differed in the preparation, composition (more organic carrier and frits of glaze), and manufacturing procedure of the CuXX component. The above pastes were used only to apply to the front electrode contacts. Other pastes used to make the back, contact were provided by the ECN part of TNO.

After the metallization process of each series, the characteristics the I-V of the solar cells were measured at the solar radiation simulator by American company Sinton. The I-V characteristics were measured for all solar cells. In other cases, cells were not measured when the first five cells in a given series had parameters with an unacceptable value. The topography and cross-section of investigated pastes of the front metallization of silicon solar cells were measured using Zeiss Supra 35 scanning electron microscope (SEM) using secondary electron detection with accelerating voltage in the range 5–20 kV. Moreover, both thickness and height of front metallization were determined by checking surface topography and the cross-section measured using the same scanning electron microscope as was mentioned before.

## 4. Silicon Solar Cells Parameters

The observations in the scanning electron microscopy provided insights on the morphology of the front electrode deposited from the A paste, B paste, and REF paste after co-fired process in the infrared furnace. The SEM results showed a porous structure (Figure 7a,c,e,g,i,k). Moreover, obtained electrodes demonstrated a structure with similar density. The thickness measurement results of the front electrodes obtained after co-firing process in the furnace on silicon substrates were the following: 40 µm (paste A), 45 µm (paste B), and 50 µm (REF paste). This difference mainly resulted from the composition of the paste and its rheological properties. Based on the fractographic investigations, it was found that front metallization obtained from the experimental and commercial REF pastes by the co-fired process in the IR furnace demonstrated connection with substrate without defects and delaminations. The electrode layer created many homogenous connections with the silicon substrate, which are close to continuous connection (Figure 7b,d,f,h,j,l). Moreover, comparison of height of the front electrodes deposited from experimental and commercial paste after metallization process amounts to 11 µm difference. It was found that among analyzed front metallization the best structural properties were exhibited by paste A. In the case of this front electrode co-fired in the furnace, it was found that the compound structure contained numerous cracks on the entire electrode surface (Figure 7b) which were not found for the REF. Moreover, this electrode demonstrated a porous structure similar to those obtained as a result of co-firing with the standard Ag paste, and their porosity grade depended on co-firing temperature, similar as in the case of the commercial paste. Thus, the paste with the XX component resembles very well REF paste confirming successful implementation of the copper volumetric filler into the Ag paste.

The fabricated solar cells were characterized by the current-voltage (I-V) measurements using a solar radiation simulator (Sinton FCT-450: Light I-V Testing for Solar Cells). The fabricated solar cells were co-fired at the temperature 930 and 940 °C and the I-V parameters for selected cells are summarized in Table 3 and Table 4.

In the case of the following structures, Al-BSF and PERC, the use of a metallization set temperature of 930 °C for the NPCuXX, paste B allowed the production of a cell with: FF = 72.63%, E_ff_ = 17.25%, and FF = 73.85%, and E_ff_ = 17.38%. At the metallization temperature of 920 °C, the solar cells with the electrode using paste B did not reach an FF value higher than 73.501, thus other results of paste B were not further analyzed. 

In the case of paste A, much higher values of the I-V characteristic parameters of the solar cells were obtained and further analysis of the results was focused only on the cells fabricated with NPCuXX-A and PV20A pastes. At a metallization temperature of 940 °C, the FF value of 77.92% was obtained for cell No. A-2 (Table 3) from paste A on the Al-BSF structure, a value lower by only 1.623% compared to the reference cell No. REF-2 made of commercial Du Pont PV20A Ag paste. While maintaining the metallization temperature of 940 °C but lowering the temperature in the preceding zone from 795 to 775 °C, so by 20 °C, there was a slight decrease in FF to 77.555% for cell No. A-4 (Table 3) from paste A on the Al-BSF structure. 

In the case of the PERC structure, the use of a metallization temperature of 930 °C for paste A allowed the production of a cell with FF = 77.70% and E_ff_ efficiency = 19.49% (cell No. A-5, Table 4), which is the highest value so far reported for a silicon electrode with an electrode paste prepared on the basis of massive Cu particles. Comparison of electrical parameters of selected cells with the highest FF and E_ff_ value with a front electrode applied from a modified paste A in the Al-BSF and PERC structure with the parameters of cells made from the commercial paste (Table 4) indicates that at selected metallization temperatures in the range of 930–940 °C, paste A allows to create a cell with parameters at a similar level as the reference paste.

The I-V characteristics of the cells with higher FF were numerically fitted with the double diode exponential relationship (DEM) and the results are summarized in Table 5.

The differences in the parameters are analyzed using a two-diode model. From the I-V data, the diffusion current density J_O1_ provides information about the recombination in the emitter and at the surface. The application of paste A did not result in a significantly higher recombination current (Table 5) which means, that J_O1_ is not directly responsible for the difference in V_oc_ value, about 10 mV, between solar cells with paste A and reference cells Al-BSF type (Table 3) and PERC type (Table 4) fabricated with Ag paste. The difference in V_oc_ is also not monitored in the effective lifetime results (Table 5). The solar cells with paste A have comparable values of the series resistance R_s_ (Table 3 and Table 4) due to the not optimized components for a new NPCuXX paste, especially glass frit composition**.** It can be seen in Figure 7, that after metallization process, the Ag paste poses higher densification between metal powders than paste A. The resulting increase in R_s_ results in a slight but constant decrease in the current density, which consequently also leads to a constant decrease in the V_oc_ value for cells made of paste A. The J_O2_ is considered a measure for recombination in the space charge region and is related to the density of recombination centers in solar cells. The calculated J_O2_ values (Table 5) did not show the degradation effect, which suggests a lack of Cu impurity under the electrode, i.e., in the emitter region.

The most important parameter opening the possibility of providing effective application for modified NPCuXX paste is the fill factor, series, and shunt resistance characterizing the produced solar cells. Equally important is the V_oc_ which is at an alarmingly low level of 630 mV. For this reason, the completed fabrication process of the cells with modified NPCuXX paste and Ag paste resulted in a cell efficiency as high as 19.49% calculated from parameters values: V_oc_ = 0.65 V, I_sc_ = 9.48 A, and FF = 77.70% solar cell PERC type and 18.68% calculated from parameters values: V_oc_ = 0.63V, I_sc_ = 9.26 A, and FF = 77.92% solar cell Al-BSF type. The most important results in table (Table 5) is the J_O1_ and J_O2_ of cells with A and B paste contacts with respect to the reference Ag paste. These results show that the CuXX component properly blocks the diffusion of Cu atoms into the p–n junction region of the solar cell.

## 5. Conclusions

The CuXX component and NPCuXX paste were elaborated and tested for front electrode formation of Al-BSF and PERC solar cells. The paper presented for CuXX material a highest fill factor of 77.92% and 77.69% for mentioned structures, respectively. The CuXX component was prepared by an industrially-viable chemical method. It is the first presented paste containing the copper volumetric material which allows to fabricate front contact of the silicon solar cells by high temperature metallization process. Based on electrical and metallographic observations of front metallization, it was found that the obtained solar cells showed parameters comparable to the ones obtained with pure Ag commercial paste. The measurement results of J_01_ and J_02_ for cells with NPCuXX pastes show that the CuXX component properly blocks the diffusion of Cu atoms into the p-n junction and with optimal firing temperatures, the cells achieve electrical parameters close to the cells manufactured with the Ag paste. There is room for further optimization of this technology, however this requires the use of a full-scale production facility, which calls for cooperation with industrial partners.

## Figures and Tables

**Figure 1 materials-11-02493-f001:**
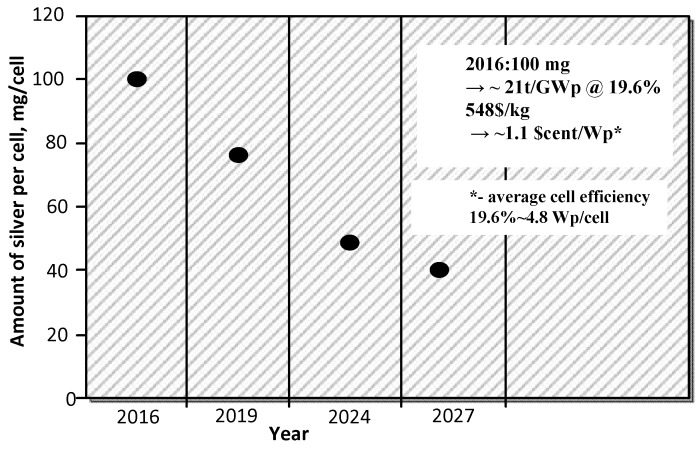
Prediction of the reduction in the use of Ag in paste applied for contact deposition [5].

**Figure 2 materials-11-02493-f002:**
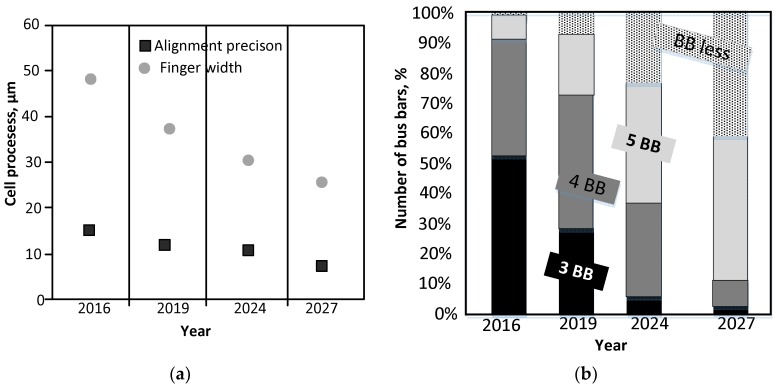
The data of the front line and bus bars electrode concerning the manufacturing process of the Si crystalline solar cells: (**a**) finger width, (**b**) number of bus bars [5].

**Figure 3 materials-11-02493-f003:**
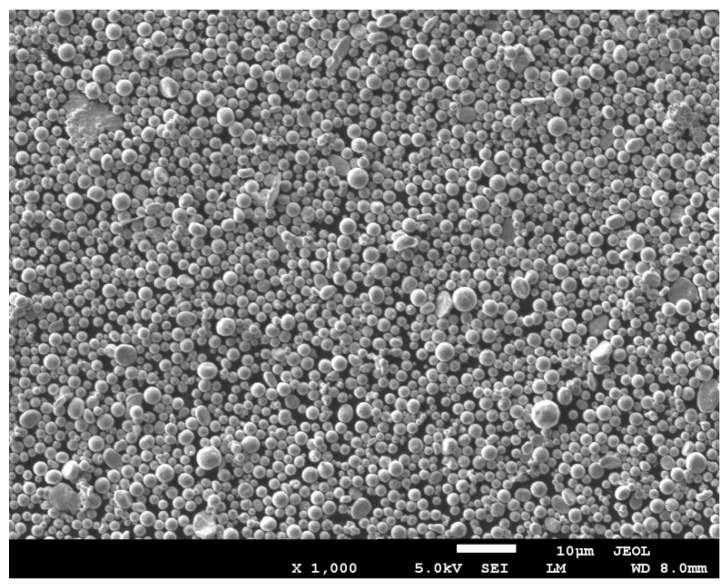
Scanning electron microscopy (SEM, Supra 35, Zeiss manufactuer, Gliwice, Poland) image of the copper grains used for the CuXX component fabrication.

**Figure 4 materials-11-02493-f004:**
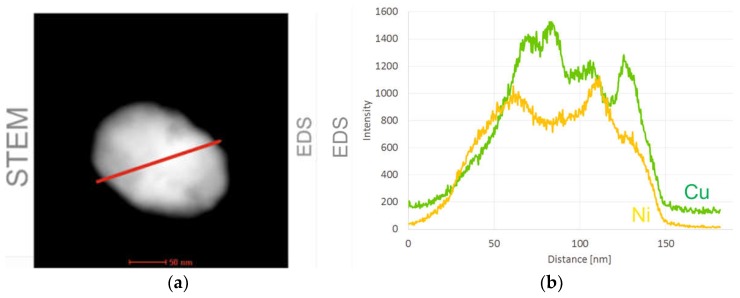
The component grain of CuXX illustrated by scanning transmission electron microscope method (STEM, Tecnai G2 F20, FEI, Krakow, Poland) (**a**), and analysis of content distribution of Cu and Ni on cut line (red line in STEM picture) (**b**).

**Figure 5 materials-11-02493-f005:**
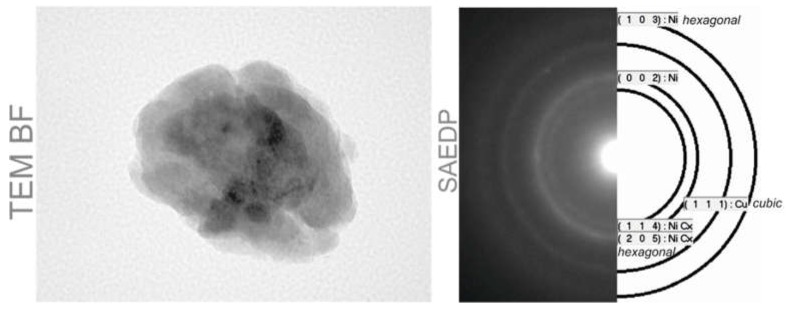
Image of the grain component CuXX performed in spectroscopy TEM method and diffraction image of the grain with assigned reflexes.

**Figure 6 materials-11-02493-f006:**
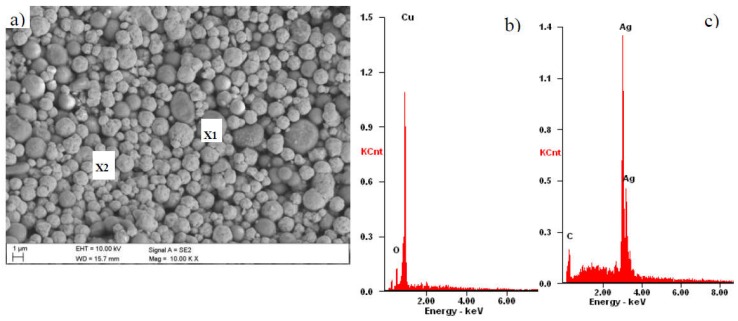
SEM micrograph presenting experimental NPCuXX pastes: (**a**) A, (**d**) B and commercial, and (**g**) REF (reference) paste, and plot of energy dispersion X-rays shown in (**b**–**h**) from micro-areas in insets from (**a**–**g**), where X1 and X2 are micro-areas which were performed energy dispersive spectroscopy investigations.

**Figure 7 materials-11-02493-f007:**
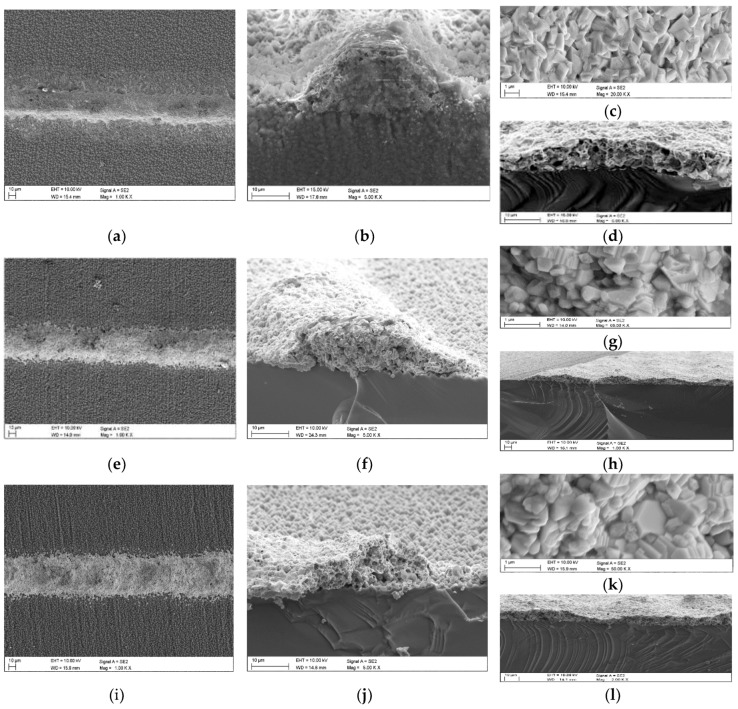
Surface layer topography made from paste A (**a**,**c**), paste B (**e**,**g**), and REF paste (**i**,**k**) on silicon surface and deposited by screen printing method and metallized in an IR furnace. Fracture of front electrode from paste A (**b**,**d**), paste B (**f**,**h**), and REF paste (**j**,**l**) obtained by SEM method.

**Table 1 materials-11-02493-t001:** Components of industrial deposited pastes [13,14].

Components	wt.%	Literature
Silver powder	70–85	[14,15,16]
Glass frits (composition); PbO—86%, B_2_O_3_—11%, SiO_2_—3% PbO, SiO_2_, B_2_O_3_, Al_2_O_3_, CaO, ZnO; PbO, ZnO, B_2_O_3_, Bi_2_O_3_; PbO, SiO_2_, ZnO, Al_2_O_3_, B_2_O_5_; SiO_2_	0–5 3 - - - 2	[16,17,18,19,20,21]
Cellulosic resin: ethyl cellulose ethoce	3–15	[22]
Solvent: solvent with small molecular weight solvent	3–15 - -	[14,23]
Additives (rheological modifiers and surfactants): ethyl cellulose+ terpineol; ethyl cellulose *	0–2 16 -	[14,24]

* Mixtures with different BCA (butyl cyanoacrylate)-to-xylene weight ratios, (-) lack of % detailed information.

**Table 2 materials-11-02493-t002:** Detailed properties of NPCuXX paste deposited onto metallic contacts.

Technical Parameter	Value and Size of the Parameter
The content of CuXX particles	>50% by weight
Addition: silver paste	yes
Application technique	screen printing, pattern
The force of adhesion of the soldered connection strip to a 2 mm wide path	>1 N
Possible path height and width	20–25 μm, 40–80 μm
It is possible to create a cell with specific serial resistance	<1 Ω·cm^2^
Resistivity	<4 mΩ·cm
Viscosity of the paste	260–420 Pa·s
The possibility of getting a contact to the type emitter	50–100 Ω/square
Additives	glazes, oxides, organic thinners, other *
Guarantee period	6 months
Storage temperature	5–25 °C
Deposition and print temperature	15–25 °C
Thixotropy of the paste	the paste should be mixed in the rolling process at speeds below 30 rpm and during 12–48 h with temperature 5–35 °C
Co-firing temperature	over 700 °C

Contractor assure of composition of experimental paste about the weight 1000 g on the basis of experimental results and literature date: powders (CuXX—600 g and Ag—250 g) 850 g, organic carrier 100 g (light and resin), frits of glaze 50 g. Diameter of CuXX in range from 1 to 2 μm and Ag from 0.5 to 1.5 μm.

**Table 3 materials-11-02493-t003:** The current-voltage (I-V) parameters of the 244 cm^2^, Cz-Si, Al-BSF solar cells area of 244 cm^2^ fabricated by a screen-printing technique and co-fired at 940 °C.

Cell Number		I_sc_ (A)	V_oc_ (V)	I_m_ (A)	V_m_ (V)	P_m_ (W)	FF (%)	E_ff_ (%)	R_sh_ (Ω)	R_s_ (Ω)
**PV20A paste**										
REF solar cells	REF-1	9.360	0.641	8.745	0.545	4.767	79.384	19.537	29.16	0.0025
	REF-2	9.397	0.643	8.786	0.547	4.808	79.541	19.705	50.40	0.0025
	REF-3	9.406	0.643	8.776	0.548	4.809	79.469	19.712	35.46	0.0025
**NPCuXX-paste A**										
Solar cells from A	A-1	9.233	0.632	8.554	0.530	4.534	77.690	18.582	30.54	0.0037
	A-2	9.256	0.631	8.636	0.527	4.557	77.918	18.679	34.91	0.0038
	A-3	9.278	0.632	8.648	0.524	4.534	77.267	18.585	42.83	0.0043
	A-4	9.256	0.632	8.551	0.531	4.543	77.557	18.621	34.39	0.0039

Where: I_sc_—short circuit current, V_oc_—open circuit voltage, I_m_—current in optimum power point, V_m_—voltage in optimum power point, P_m_—optimum power point, FF—fill factor, E_ff_—conversion efficiency, R_s_—specific series resistance, R_sh_—specific shunt resistance.

**Table 4 materials-11-02493-t004:** The current-voltage (I-V) parameters of the 244 cm^2^, Cz-Si, PERC solar cells area of 244 cm^2^ fabricated by application of the NPCuXX paste for front electrode by a screen-printing technique and co-fired at 930 °C and 940 °C.

Cell Number on the ECN List		I_sc_ (A)	V_oc_ (V)	I_m_ (A)	V_m_ (V)	P_m_ (W)	FF (%)	E_ff_ (%)	R_sh_ (Ω)	R_s_ (Ω)
**PV20A paste**										
REF solar cells	REF-4 (940 °C)	9.579	0.655	8.953	0.548	4.912	78.266	20.132	102.47	0.0037
	REF-5 (930 °C)	9.560	0.658	8.898	0.552	4.917	78.151	20.153	129.68	0.0038
**NPCuXX-A paste**										
Solar cells from A	A-5 (930 °C)	9.484	0.645	8.813	0.539	4.756	77.697	19.491	36.04	0.0039
	A-6 (930 °C)	9.474	0.645	8.807	0.538	4.744	77.517	19.443	68.59	0.0042

**Table 5 materials-11-02493-t005:** Selected electrical parameters of the Cz-Si based solar cells with the highest FF and E_ff_ values with the front electrode applied from the modified NPCuXX paste produced on the ECN part of TNO line in the structure of Al-BSF and PERC compared to the cells produced using the reference paste.

Investigated Paste	Cell Number	J_01_ (A/cm^2^)	J_02_ (A/cm^2^)	τ (μs)
**Solar cell Al-BSF type**				
PV20A	REF-1	4.62 × 10^−13^	1.91 × 10^−9^	101.94
	REF-2	6.18 × 10^−13^	4.84 × 10^−9^	69.72
	REF-3	5.99 × 10^−13^	4.56 × 10^−9^	73.01
NPCuXX-A	A-1	5.82 × 10^−13^	3.86 × 10^−9^	76.88
	A-2	8.46 × 10^−13^	6.61 × 10^−9^	51.76
	A-3	5.73 × 10^−13^	4.93 × 10^−9^	75.33
	A-4	8.43 × 10^−13^	3.50 × 10^−9^	55.22
**Solar cell PERC type**				
PV20A	REF-4	8.19 × 10^−13^	5.17 × 10^−9^	55.29
	REF-5	7.40 × 10^−13^	2.20 × 10^−9^	66.40
NPCuXX-A	A-5	3.58 × 10^−13^	2.95 × 10^−9^	126.99
	A-6	5.47 × 10^−13^	2.73 × 10^−9^	84.19

Where: J_01_—diffusion component of dark current, J_02_—generation-recombination component of dark current, τ—effective lifetime of the load carriers.

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
