# Peer review of "Copper-Based Volumetric Filler Dedicated for Ag Paste for Depositing the Front Electrodes by Printing on Solar Si Cells"

_materials, 2018, doi:10.3390/ma11122493_

Round 1

Reviewer 1 Report

This paper reports a new paste based on copper composite (NPCuXX) for fabrication front electrodes in silicon solar cells. The NPCuXX paste has been applied both to conventional cell structures as Al-BSF and PERC and finally solar cells with front electrode deposited by screen-printing method were fabricated and characterized by current-voltage techniques. The Si solar cells with a highest fill factor of 77.69 %-77.92% were obtained. This study is interested to the readers. However, it is need some improvements:

1. In Keywords, the “front side metallization” may be revised as “front electrode”, because the word “front side metallization” is not presented on Title and Abstract of the submitted manuscript.

2. Page 5 of 12, in Table 2, the authors claimed that “Resistivity < 4 m Ω/□”. Please check the correctness of the unit of resistivity.

3. Page 5of 12 and 6 of 12, please check the correctness of symbol of “÷” in Table 2.

4. Page 6 of 12 and 7 of 12, please give the quantity name and its unit in y-axis in Figs. 6 (b), (c), (e), (f), and (h) for EDS investigations.

5. Page 9 of 12, the authors claimed that “In the case of the following structures Al-BSF and PERC, the use of a metallization set temperature of 930°C for the NPCuXX - B paste allows the production of a cell with: FF = 72.63, Eff = 17.25% and FF=73.85, Eff =17.38 %. At the metallization temperature of 920 °C, the solar cells with the electrode using B paste do not reach the FF value higher than 73.501, thus other results of B paste were not further analyzed”. Please give the symbol of “%” after the magnitude of FF.   

6. Please give the area information of the fabricated silicon solar cells for readers to evaluate photovoltaic performance in Table 3 and Table 4.

7. The authors claimed that “In this work we present research results on a new paste NPCuXX based on copper composite (CuXX) for fabrication front electrodes in silicon solar cells”. Can authors give the mean of XX or more specific information for the readers to understand in the revised manuscript?

8. The authors should be revised the format of references according to Materials journal.

Author Response

1. In Keywords, the “front side metallization” may be revised as “front electrode”, because the word “front side metallization” is not presented on Title and Abstract of the submitted manuscript.

Response: We change in the in keywords: from the “front side metallization” into “front electrode”.

2. Page 5 of 12, in Table 2, the authors claimed that “Resistivity < 4 m Ω/□”. Please check the correctness of the unit of resistivity.

Response: It should be the unit mWcm2.

3. Page 5 of 12 and 6 of 12, please check the correctness of symbol of “÷” in Table 2.

Response: We change the symbol from “÷” to “-“in Table 2.

4. Page 6 of 12 and 7 of 12, please give the quantity name and its unit in y-axis in Figs. 6 (b), (c), (e), (f), and (h) for EDS investigations.

Response: The quantity name and its unit in y-axis are in red color. We have changed the size of the figures.

5. Page 9 of 12, the authors claimed that “In the case of the following structures Al-BSF and PERC, the use of a metallization set temperature of 930°C for the NPCuXX - B paste allows the production of a cell with: FF = 72.63, Eff = 17.25% and FF=73.85, Eff =17.38 %. At the metallization temperature of 920 °C, the solar cells with the electrode using B paste do not reach the FF value higher than 73.501, thus other results of B paste were not further analyzed”. Please give the symbol of “%” after the magnitude of FF.

Response: The authors completed in% at FF.

6. Please give the area information of the fabricated silicon solar cells for readers to evaluate photovoltaic performance in Table 3 and Table 4.

Response: We write the area 244 cm2 into the text.

7. The authors claimed that “In this work we present research results on a new paste NPCuXX based on copper composite (CuXX) for fabrication front electrodes in silicon solar cells”. Can authors give the mean of XX or more specific information for the readers to understand in the revised manuscript?

Response: A detailed explanation of the composition and method of manufacturing the CuXX component is described in patent application [12]. The authors have completed this position in abstract.

[12] Patent application No. EP18190412.9. A method for manufacturing modified electrically-conductive copper particles and modified electrically-conductive copper particles.

8. The authors should be revised the format of references according to Materials journal.

Response: The authors revised the format of references according to Materials journal. We shorter the references because there were same mistakes.

[1] Goodrich, A.;  Hacke, P.; Wang, Q.; Sopori, B.; Margolis, R.; James, T.L.; Woodhouse, M. A wafer-based monocrystalline silicon photovoltaics road map: Utilizing known technology improvement opportunities for further reductions in manufacturing costs. Solar Energy Materials & Solar Cells 2013, 114, 110–35.

[2] Green, M.A.; Hishikawa, Y.; Dunlop, E.D.; Levi, D.H.; Hohl-Ebinger, J.; Ho-Baillie, A.W.Y. Solar cell efficiency tables (version 51). Prog Photovolt Res Appl. 2018, 26, 3–12.

[3] Dobrzański, L.A.; Musztyfaga, M. Effect of the front electrode metallisation process on electrical parameters of a silicon solar cell, Journal of Achie­vements in Materials and Manufacturing Engineering 2011, 48 /2,  2, 115-144.

[4] Rudolph, D.; Olibet, S.; Hoornstra, J.; Weeber, A.; Cabrera, E.; Carr, A.; Koppes, M.; Kopecek, R. Replacement of silver in silicon solar cell metallization pastes containing a highly reactive glass frit: Is it possible? Energy Procedia 2013, 43, 44-53.

[5] Metz, A.; Fischer, M.; Trube, J. International Technology Roadmap for Photovoltaics (ITRPV) 8th edition: Crystalline silicon Technology-Current Status and Outlook, PV Manufacturing in Europe Conference, Brussels, May 19th 2017.

[6] https://zloto.bullionvault.pl/wykres-cen-zlota.do (archived on 20 September 2018)

[7] Ralph, E.L. Recent advancements in low cost solar cell processing, Proc. 11th IEEE PVSC Scottsdale, Arizona, USA, 1975.

[8] Berwind J. PV manufacturing materials: Technological and process-related options for cost reduction, Photovoltaics International 2012, 15, p. 36 – 49.

[9] Schuler, S.; Luck, I.  Cell metallization by screen printing: Cost, limits and alternatives; Photovoltaics Intenational 2014, 23.

[10] pvinsight.com (archived on 20 September 2018)

[11] https://pv-system.pl/blog/do-2025r-energia-sloneczna-bedzie-tansza-niz-z-wegla-lub-gazu (archived on 22 September 2018)

[12] Patent application No. EP18190412.9. A method for manufacturing modified electrically-conductive copper particles and modified electrically-conductive copper particles.

[13] Caballero, L.J. Contact Definition in industrial silicon solar cells. Solar Energy 2006, 375-398.

[14] Bhosale1, D.; Wagh, M.; Shinde, N. Review on front contact metallization paste using silver nano particles for performance improvement in polycrystalline silicon solar cell International, Journal of Scientific Engineering and Applied Science 2015, 1, 2, p. 58-62.

[15] Dobrzański, L.A. Musztyfaga, M. Effect of the front electrode metallisation process on electrical parameters of a silicon solar cell, Journal of Achie­vements in Materials and Manufacturing Engineering 2011, 48 /2, 2, 115-144.

[16] Mette, A. New Concepts for Front Side Metallization of Industrial Silicon Solar Cells. Dissertation zur Erlangung des Doktorgrades der Fakultät für Angewandte Wissenschaften der Albert-Ludwigs-Universität Freiburg im Breisgau, Fraunhofer-Institut für Solare Energiesysteme Freiburg im Breisgau, 2007.

[17] Rudolph, D. Olibet, S.; Hoornstra, J.; Weeber, A.; Cabrera, E.; Carr, A.; Koppes, M.; Kopecek, R. Replacement of silver in silicon solar cell metallization pastes containing a highly reactive glass frit: Is it possible?. Energy Procedia 2013, 43, 44 – 53.

[18] Hong, K-K. Mechanism for the formation of Ag crystallites in the Ag thick-film contacts of crystalline Si solar cells, Solar Energy Mater. Solar Cells 2009, 93, p. 898-904.

[19] Hoenig, R. Impact of screen printing silver paste components on the space charge region recombination losses of industrial silicon solar cells. Solar Energy Mater. Solar Cells 2012, 106, p. 7-10.

[20] Panek, P.; Socha. R.; Putynkowski, G.; Slaoui A. The new copper composite of pastes for Si solar cells front electrode application, Energy Procedia 2016, 92, p. 962-970.

[21] Musztyfaga-Staszuk, M. SLS: One of the Modern Technologies of Laser Surface Treatment. International Journal of Thermophysics 2017,  38:130, DOI 10.1007/s10765-017-2263-1.

[22] Yu-Shun, C.; Chin-Lung, C.; Thou-Jen, W.; Chi-Cheng, C.  Development of Screen-Printed Texture-Barrier Paste for Single-Side Texturization of Interdigitated Back-Contact Silicon Solar Cell Applications. Materials 2013, 6, 4565-4573; doi:10.3390/ma6104565

[23] Ching‐his, L.; Shih‐Peng, H.; Wei‐Chih H. Silicon Solar Cells: Structural Properties of Ag‐ Contacts/Si‐Substrate, In book: Solar Cells - Silicon Wafer-Based Technologies Published: November 2nd 2011, 978-953-307-747-5.

[24] Dong-Youn, S.; Jun-Young, S.; Hyowon, T.; Doyoung, B. Bimodally dispersed silver paste for the metallization of a crystalline silicon solar cell using electrohydrodynamic jet printing. Solar Energy Materials & Solar Cells 2015, 136, 148–156.

Important think:

Because in paper was a mistake:

Małgorzata Musztyfaga-Staszuk 1,*, Grzegorz Putynkowski 2, Robert Socha 3, Maciej Stodolny 4, Panek Piotr

We also change into (red color also in paper)

Małgorzata Musztyfaga-Staszuk 1,*, Grzegorz Putynkowski 2, Robert Socha 3, Maciej Stodolny 4, Piotr Panek

Reviewer 2 Report

Comments are attached.

Author Response

The manuscript investigates the use of Ag paste to be mixed with a high percentage of Cu, Cu being introduced to contribute in reduction of the manufacturing cost of Si photovoltaics. The results are interesting and worth publishing in Materials, however, the following concerns should be considered, and a revision would be required.

1. The manuscript does not provide enough information on previous efforts focused on use of new paste formulations for front metallization for c-Si the first presented paste containing the copper volumetric material which allows to fabricate front contact of the silicon solar cells by high temperature metallization -Ag mixture or the first report of Cu-Ag for high temperature metallization, considering no information on previous efforts is provided.

Response: In the first works on the development of the paste [20], results were presented in which the cells had an FF of 73%. Subsequent work concerned the study of the parameters of CuXX component molecules and the effect of NPCuXX paste on cell parameters [25, 26]. The refining of the paste's method and ingredients as well as the creation of links on the ECN line made it possible to achieve such spectacular results.

The authors supplemented this information in the text.

[20] Panek, P.; Socha. R.; Putynkowski, G.; Slaoui A. The new copper composite of pastes for Si solar cells front electrode application, Energy Procedia 2016, 92, p. 962-970.

[25] Musztyfaga-Staszuk, M.; Woźny, K.; Putynkowski, G.; Zięba, P.; Panek, P.; Marynowski, P. The properties of the developed technology for production of copper component and paste used in the production process of electrical contacts of silicon cells, METAL 2017: 26th International Conference on Metallurgy and Materials, Brno, Czechy, p. 1983-1989, ISBN: 978-80-87294-79-6.

 [26] Musztyfaga-Staszuk, M.; Major, Ł.; Putynkowski, G.;  Sypień, A.; Gawlińska, K.; Panek. P.; Zięba. P.; New kind of Cu based paste for Si solar cells front contact formation, 2018, Materials Science-Poland, DOI: 10.2478/msp-2018-0068.

2. The new paste is labelled as NPCuXX. What does NP stand for? And XX might just be Ni.

Explanation: NPCuXX (where: NP- new paste, CuXX – component, XX – a modifier consist which Ni and other important elements).

The authors included this explanation in the abstract.

3. The I-V characteristic curves should be presented.

Response: Solar cells were measured at the ECN facility in the Netherlands. The measurement data included in the EXCEL sheets does not include graphical presentation of results.

4. In line 258, the authors mention that the solar cells based on paste A formulation have lower series resistance. The table 4 and 5 show otherwise and should be corrected.

Response: The authors change the sentence from:

The solar cells with A paste have slightly lower values of the series resistance Rs (Table 3 and 4) due to the not optimized components for a new NPCuXX paste, especially glass frit composition.

onto:

The solar cells with A paste have comparable values of the series resistance Rs (Table 3 and 4) due to the not optimized components for a new NPCuXX paste, especially glass frit composition.

5. In line 142, the authors mentioned that the comparison between the tables 1 and 2 shows that the new NPCuXX paste is not inferior to the quality of commercial pastes. However,

the table 1 does not provide the required parameters mentioned in table 2 to make comparisons.

Response: Du Pont does not provide comprehensive information about the paste parameters and does not allow the publication of detailed test results. NPCuXX paste is currently produced on an experimental line and its description in the work contains only the data allowed by the manufacturer.

6. Paste A formulation worked while B did not for the solar cells fabrication. However, no proper reasonings were provided for the result. In addition, the authors should also mention what is the exact difference between paste A and B and what was the objective of studying two different pastes.

Response: The pastes A and B differed in the preparation, composition (more organic carrier and frits of glaze) and manufacturing procedure of the CuXX component.

The authors have completed this information in the text

Round  2

Reviewer 1 Report

This paper reports a new paste based on copper composite (NPCuXX) for fabrication front electrodes in silicon solar cells. The NPCuXX paste has been applied both to conventional cell structures as Al-BSF and PERC and finally solar cells with front electrode deposited by screen-printing method were fabricated and characterized by current-voltage techniques. The Si solar cells with a highest fill factor of 77.69 %-77.92% were obtained. This study is interested to the readers. However, it is need some improvements:

1. Pages 5 of 12 and 6 of 12, in Table 1 and Table 2, the authors claimed that “Resistivity < 4 m Ωcm2”. Please check again the correctness of the unit of resistivity. In general, the resistivity unit is “Ω-cm”.

2. The graphs of Figs. 6 (b), (c), (e), (f), and (h) for EDS investigations are need to improve clearly for readers to evaluate the quality of presentation.

Author Response

1. Pages 5 of 12 and 6 of 12, in Table 1 and Table 2, the authors claimed that “Resistivity < 4 m Ωcm2”. Please check again the correctness of the unit of resistivity. In general, the resistivity unit is “Ω-cm”.

Response: Yes it is true, the unit of resistivity should be Ω-cm. It is our mistake. I apologize for this. 

2. The graphs of Figs. 6 (b), (c), (e), (f), and (h) for EDS investigations are need to improve clearly for readers to evaluate the quality of presentation.

Response: We have improved the units on the ranks so that they are easier to read.

Reviewer 2 Report

The authors have addressed the comments.

Author Response

Dear Sir/Madam, The article was sent for the English correction. correction.